# *Lacticaseibacillus rhamnosus* Probio-M9-Driven Mouse Mammary Tumor-Inhibitory Effect Is Accompanied by Modulation of Host Gut Microbiota, Immunity, and Serum Metabolome

**DOI:** 10.3390/nu15010005

**Published:** 2022-12-20

**Authors:** Weiqin Zhang, Yong Zhang, Yalin Li, Da Ma, Heping Zhang, Lai-Yu Kwok

**Affiliations:** 1Inner Mongolia Key Laboratory of Dairy Biotechnology and Engineering, Inner Mongolia Agricultural University, Hohhot 010018, China; 2Key Laboratory of Dairy Products Processing, Ministry of Agriculture and Rural Affairs, Inner Mongolia Agricultural University, Hohhot 010018, China; 3Key Laboratory of Dairy Biotechnology and Engineering, Ministry of Education, Inner Mongolia Agricultural University, Hohhot 010018, China; 4School of Chemistry and Biological Engineering, University of Science and Technology Beijing, Beijing 100024, China

**Keywords:** breast cancer, mammary tumor, *Lacticaseibacillus rhamnosus* Probio-M9, gut microbiota, serum metabolite, immunity

## Abstract

Gut microbiome may influence tumor growth and cancer treatment efficacy, so it is a potential target for tumor prevention/treatment. This pilot study investigated the preventive and therapeutic effects of a probiotic strain, *Lacticaseibacillus rhamnosus* Probio-M9 (Probio-M9), against murine mammary cancer. Thirty-six female mice were randomly divided into three groups (*n* = 12 per group): control (without tumor transplantation), model (tumor transplantation; no probiotic administration), and probiotic (30-day oral gavage of probiotic, started seven days before tumor transplantation). Changes in tumor size were recorded, and blood, tumor tissue, and stool samples were collected at the end of the trial for analyses. Comparing with the model group, the probiotic group had a significantly smaller tumor volume (*p* < 0.05), a higher fecal microbiota Shannon diversity index, with significant modifications in the gut microbiota structure (*p* < 0.05), characterized by more *Alistipes* sp._2, *Porphyromonadaceae* bacterium_7, and *Bacteroidales* bacterium 55_9 (*p* < 0.05). Additionally, Probio-M9 administration elevated the serum IFN-γ, IL-9, IL-13, and IL-27 levels and several metabolites (e.g., pyridoxal, nicotinic acid, 3-hydroxybutyric acid, glutamine; *p* < 0.05), while reducing IL-5 (*p* < 0.05). These changes might be associated with the protective effect of Probio-M9 against mammary tumor growth. Thus, probiotic administration could harness host gut microbiome in anti-cancer responses.

## 1. Introduction

The World Health Organization states that “Cancer is a leading cause of death worldwide, accounting for nearly 10 million deaths in 2020, or nearly one in six deaths.” [1]. In recent years, breast cancer has become one of the most common cancers in women and the most common cancer overall [2]. Approximately 15–20% cancer cases worldwide are notably driven by microbial pathogenesis, and an even greater number of malignancies are attributed to gut dysbiosis [3]. It has been reported that brain cancer, breast cancer, lung cancer, and other tumor samples comprise tumor type-specific intracellular bacteria, localizing within both tumor and immune cells [4]. Thus, the intra-tumor mass environment could be considered a complicated and dynamic microecosystem, regulated by the tripartite interaction between the host immunity, tumor-associated microbiota, and tumor [5].

The human colon environment is inhabited by a large population of gut microbiota that co-evolves with humans. The gut microbiota is involved in physiological homeostasis, including the ability of reducing systemic inflammation and shaping the innate and adaptive immunity [6,7]. There is a delicate symbiotic balance between the host immunity and gut microbiome. However, should such a balance state be disturbed, gut microbial dysbiosis and subsequent pathological processes could occur. For example, gut microbes may play a role in the occurrence and development of cancer through their effects on cell signaling pathways via metabolite production and via modulation of the host immune state [8]. Recent progresses in molecular biology have enabled extensive studies of cancer pathogenesis in relation to gut microbes. Several studies found that elevated levels of endogenous or circulating estrogen are directly associated with an increased risk of breast cancer, as estrogen promotes the proliferation of normal breast epithelium and cancer cells, as well as breast cancer metastasis through Notch signaling [9,10]. There is a bidirectional interaction between the gut microbiome and the endocrine system. The gut microbiome produces a variety of hormone-like substances related to metabolism; meanwhile the endocrine reacts to fluctuation in the gut microbiome. Especially, women with high estrogen levels had a more diverse gut microbiome [11,12]. A smaller fecal microbiota diversity and altered fecal microbiota composition were observed in postmenopausal women with newly diagnosed breast cancer compared with similar healthy women, and these cancer subjects had elevated urinary estrogen levels, implicating a possible role of gut microbiome in breast cancer [13]. Short-chain fatty acids (SCFAs) are another group of important gut microbiota metabolites that can act directly on various intestinal immune cells [14]. The levels of colonic SCFAs in premenopausal breast cancer patients decreased significantly, suggesting a key role of SCFAs in the pathological mechanism of premenopausal breast cancer [15].

Probiotics are defined as “live microorganisms which when administered in adequate amounts confer a health benefit on the host” [16]. Many studies have outlined the beneficial effects of probiotics on various ailments, such as irritable bowel syndrome, ulcerative colitis, and constipation [17,18,19]. Previous studies have shown that probiotics can strengthen host immunity, including antitumor responses, prolonging the survival time in tumor-transplanted mice with or without receiving cancer treatment [20,21]. Some gut microbiota and the probiotic species, *Bifidobacterium breve*, have been found to stimulate the gut production of FOXP3^+^ regulatory T cells and T regulatory type 1 cells through the TLR2/MyD88 pathway, and the latter directly suppresses T helper (Th) 17 cells and inhibits tumor growth by secreting interleukin (IL)-10 [22,23]. Administering *Lactiplantibacillus plantatum* LS/07 repressed the tumor frequency, accompanied by an increase in CD4^+^ T-cells in tumor tissue, reduction in the serum tumor necrosis factor-α concentration, and elevation in CD8^+^ T-cell number in tumor tissue but decreased blood CD8^+^ T-cell count [24]. Regular consumption of *Lacticaseibacillus casei* Shirota and soy isoflavones since adolescence has been found to reduce breast cancer risk in Japanese women [25]. Probiotics also show great potential in other cancer clinical studies, e.g., alleviating post-operative complications in colon cancer patients, relieving therapy-related toxicity, and improving the quality of life [26].

*Lacticaseibacillus rhamnosus* Probio-M9 (Probio-M9) is an novel probiotic strain isolated from human breast milk of a healthy woman, which has been shown to suppress tumor formation in the large intestine via regulating the intestinal environment and inflammation [27]. Immune checkpoint inhibitor-based immunotherapy has shown good clinical effects [28]. Although immunotherapy brings new hope to patients, it has limitations, such as unresponsiveness to treatment. Immunotherapies may also have serious side effects, such as fatigue, pruritus, vitiligo, diarrhea, and colitis [28]. Our previous study showed that full-blown anti-tumor effect of immune checkpoint inhibitors required a relatively intact host gut microbiota, and administering Probio-M9 could synergize with inflammatory mammary cancer therapy in tumor suppression in antibiotic-treated mice [29]. Provided that probiotics have the ability to modulate the host gut microbiota, and that the gut microbiota may influence the host immunity and cancer development, it may thus be of interest to investigate the antitumor effect of probiotic administration [30]. Moreover, our preliminary work in a rat model has shown that Probio-M9 could be detected in the mammary glands and mesenteric lymph nodes after oral gavage, suggesting the existence of an entero-mammary pathway through the lymphatic system. These observations together suggested that Probio-M9 could translocate to the mammary gland after ingestion and possibly confer beneficial effects in mammary tissues. Thus, this study hypothesized that Probio-M9 could also suppress mammary tumor growth via modulating host gut microbiome, immunity, and metabolism.

In this work, we aimed to investigate the anti-tumor effect of Probio-M9 in mice transplanted with mammary cancer cells. The tumor suppressive effect of Probio-M9 was evaluated by the tumor volume, gut microbiome, inflammatory factors, and serum metabolites. This work serves as a pilot study showing that probiotic administration was effective in slowing the growth of transplanted mammary tumor, and its preventive and/or therapeutic effect in anti-tumorigenesis merits further clinical investigation.

## 2. Material and Methods

### 2.1. Animals, Cancer Cell Line, and Probiotic Strain

All the protocols of animal trial in this study were approved by the Ethical Committee of Inner Mongolia Agricultural University (No. IACUC-20191117). Specific pathogen-free female BALB/c nude mice were purchased from Beijing Weitong Lihua Experimental Animal Co., Ltd. (Beijing, China). Six-week-old mice were raised and housed under specific pathogen-free conditions and received sterilized feed and water. The human breast cancer cells (cell line MDA-MB-231) were obtained from the People’s Hospital of Peking University, China. Cells were cultured in a regular CO_2_ incubator (kept at 37 °C, 5% CO_2_) with RMPI 1640 complete culture medium. Tumor xenograft was performed inside a laminar flow cabinet. The probiotic strain, Probio-M9, was provided by the Key Laboratory of Dairy Biotechnology and Engineering, Ministry of Education, Inner Mongolia Agricultural University, China.

### 2.2. Trial Design

A total of 36 mice were maintained in an animal facility (12/12 h dark/light cycle, constant temperature at 22 °C ± 1 °C). Mice were acclimatized for one week and were given standard mouse feed and water ad libitum throughout the animal study. The antitumor effect of Probio-M9 was evaluated by a murine breast tumor xenograft model. The acclimatized mice were randomized into three groups (*n* = 12 per group; Figure 1a), which were: (1) control group (without tumor xenograft or probiotic intervention); (2) model group (tumor xenograft but not probiotic intervention); and (3) probiotic group (tumor xenograft with oral gavage of probiotics). The probiotic was administered to mice by daily oral gavage (Probio-M9 at the dose of 4 × 10^9^ CFU/day) from seven days prior (day −7 to 0) to tumor inoculation (day 0) until day 22 when the trial ended. The two non-probiotic recipient groups were given an equal amount of normal saline in place of probiotic, and the control group received an equal volume of cell culture medium for tumor cell maintenance instead of tumor cells. Briefly, the skin of the inoculation site was disinfected with alcohol, and a human breast cancer cells suspension (containing 5 × 10^6^ cells) was slowly injected subcutaneously into the right axilla of all nude mice. After the injection, the skin at the inoculation site was slightly elevated for a short moment when a light pressure was applied with a sterile cotton swab. At day 22, all mice were sacrificed, and feces, serum, and tumor tissue samples were collected. Except for the tumor size measurement and histochemistry, samples from 10 mice per group (randomly chosen) were analyzed.

### 2.3. Monitoring of Tumor Growth

Changes in the tumor size of mice (*n* = 12 per group) were measured by a digital vernier calipers at days 12, 14, 16, 18, 20, and 22. With the assumption that the tumor was an ellipsoid body, the tumor volume was calculated according to the formula: V (mm^3^) = (L × W^2^)/2, where L and W were the length and width of the tumor mass [31]. At day 22, mice were sacrificed and dissected, and tumors developed in the mice were fixed.

### 2.4. Immunohistochemistry

Tumor tissues of three randomly chosen mice per group were fixed in 4% formaldehyde, embedded in paraffin, and sectioned. Immunostaining was carried out with rabbit primary antibodies (Abcam, Inc., Shanghai, China) against three different interleukins, i.e., IL-27-A, IL-5RA, and IL-9, respectively. The secondary antibody was a goat anti-rabbit horseradish peroxidase-linked immunoglobulin (Cell Signaling Technology, Inc., Shanghai, China). The DAB Horseradish Peroxidase Color Development Kit (Beyotime, Beijing, China) was used for reacting with the sections for color development. Stained sections were imaged with Leica Aperio CS2 system (Leica, Wetzlar, Germany). Positive signals of immunostaining of histological sections were quantified by using ImageJ (three mice per group, two replicates per mouse per staining). Data are presented as average optical density.

### 2.5. Fecal Sample Collection and Metagenomic Shotgun Sequencing

Feces were collected in sterile containers with the Sample Protector for RNA/DNA (Code No. 9750; Takara Biomedical Technology [Beijing] Co., Ltd., Beijing, China) and stored temporarily at −80 °C until metagenomic DNA extraction. Metagenomic DNA was extracted with the QIAGEN DNA Stool Mini-Kit (QIAGEN, Hilden, Germany) following the manufacturer’s instructions. The quality of the extracted DNA was assured by agarose gel electrophoresis and spectrophotometry (final DNA concentration >100 ng/μL; 260 nm/280 nm ratio between 1.8–2.0). Sequencing libraries were prepared by using the NEBNext^®^ Ultra™ DNA Library Prep Kit (New England BioLabs, Ipswich, MA, USA) following the manufacturer’s recommendations. Qualified DNA was subjected to shotgun metagenomic sequencing using an Illumina HiSeq 2500 instrument (generating ~5.5 ± 1.49 Gbp raw data per sample; range = 2.89 to 7.69 Gbp). Raw metagenomic reads were then processed through the KneadData quality control pipeline.

Reads of each sample were assembled into contigs using MEGAHIT [32], and contigs larger than 2000 bp were chosen for binning to gain metagenome-assembled genomes (MAGs) using MetaBAT2 [33], VAMB [34], and DAS Tool with default options [35]. The completeness and contamination of MAGs were evaluated through CheckM (https://github.com/Ecogenomics/CheckM, accessed on 15 November 2022). These MAGs were divided into different quality levels and clustered into species-level genome bins (SGBs) according to methods described in a previous work [36]. The cut-off levels of MAGs were: high-quality (completeness ≥80%, contamination ≤5%), medium-quality (completeness ≥70%, contamination ≤5%), and partial-quality (completeness ≥50%, contamination ≤5%); and high-quality genomes were clustered, and the most representative genomes of each replicate set were selected by dRep to identify SGBs, using the options -pa 0.95 and -sa 0.95. The SGBs were annotated by using Kraken2 and compared across the NCBI nonredundant Nucleotide Sequence Database (retrieved in 2021.11), and the relative abundance of each SGB was calculated through coverM (https://github.com/wwood/CoverM, accessed on 2 October 2021) using the option “--min-read-percent-identity 0.95 --min-covered-fraction 0.4”.

### 2.6. Serum Cytokine Analysis

Collected serum samples were stored at −20 °C until cytokine and metabolomics analyses. The concentrations of a multitude of cytokines were detected with the ProcartaPlex Multiplex Immunoassay (eBioscience, San Diego, CA, USA) by using the mouse Th1/Th2/Th9/Th17/Th22/Treg Cytokine Panel (17 plex) kit for detection of interferon (IFN)-γ, tumor necrosis factor-α, granulocyte macrophage colony-stimulating factor, IL-1β, IL-2, IL-4, IL-5, IL-6, IL9, IL-10, IL-12p70, IL-13, IL-17A, IL-18, IL-22, IL-23, and IL-27. The procedures were performed according to the manufacturer’s instructions.

### 2.7. Serum Metabolomics Analysis

Serum samples were thawed at 4 °C. Each sample (200 μL) was vortex mixed with 800 μL of methanol–water solution (4:1, *v*/*v*) for 1 min. Samples were centrifuged at 17,000× *g* for 15 min at 4 °C after standing at −20 °C for 60 min. Supernatants were collected and vacuum dried at room temperature. The dried samples were dissolved in 200 μL acetonitrile–water solution (1:1, *v*/*v*). After that, the mixtures were centrifuged and filtered, and the supernatants were collected for mass spectrophotometry analysis in both positive and negative ion modes using the Agilent 6545A QTOF (Agilent Technologies, Santa Clara, CA, USA).

The metabolomic data was evaluated by principal coordinate analysis and partial least squares-discriminant analysis. The variable importance in projection (VIP) score was used for screening differential abundant metabolites using the SIMCA-P +14.0 software (Umetrics, MKS Instruments Inc., Sweden), and the differential abundant biomarkers were manually inspected based on peak shape and signal-to-noise ratio. Differential abundant features were cross-compared with the blood exposure database (https://bloodexposome.org, accessed on 26 November 2021) for the best annotation results.

### 2.8. Statistical Analyses

All statistical analyses and data visualization were performed using the R software (v.4.0.3) and Adobe Illustrator environment. The Shannon index, principal coordinates analysis, and partial least squares-discriminant analysis were performed with R packages (vegan, optparse, and ggpubr). The adonis *p* value was generated based on 999 permutations. Wilcoxon tests were used to evaluate the statistical difference between groups. Correlation analyses among tumor growth inhibition, gut microbiota, serum cytokines and metabolites were performed using the Pearson rank correlation coefficient.

### 2.9. Data Availability Statement

Raw sequence data generated in this study are deposited at NCBI-SRA (BioProject: PRJNA821272; https://dataview.ncbi.nlm.nih.gov/object/PRJNA821272, accessed on 30 March 2022).

## 3. Results

### 3.1. Probio-M9 Administration Slowed down Tumor Growth in Mice

The body weight of the mice was monitored throughout the trial. The weight gain in the control and probiotic groups was more obvious than the model group, though no significant difference was observed at most time points (Figure 1b), suggesting that tumor transplantation inhibited the mouse weight gain to some extent, but supplementing Probio-M9 mitigated such effect. At day 22, the body weight of the probiotic group was significantly higher than the model group (mean body weight ± SEM of probiotic and model groups = 19.03 ± 0.81 g and 18.41 ± 0.77 g, respectively, *p* = 0.047; Figure 1b).

To evaluate the anti-tumor effect of probiotic administration, tumor growth (expressed in tumor volume) in the probiotic group was compared with the model group every other day between day 12 to day 22. Tumor growth in both groups showed an upward trend; however, mice in the probiotic groups had generally smaller tumor volume compared with the model group (Figure 1c). Obvious differences in the tumor volume were observed from day 14, although significant intergroup differences were only found at days 16, 20, and 22 (mean tumor size ± SEM of probiotic and model groups = 214.60 ± 16.61 mm^3^ vs. 166.05 ± 15.62 mm^3^, 453.88 ± 39.59 mm^3^ vs. 339.92 ± 26.66 mm^3^, 659.38 ± 49.19 mm^3^ vs. 496.04 ± 55.70 mm^3^; *p* = 0.046, 0.046, and 0.04, respectively; Figure 1d). Indeed, even at days 14 and 18, especially at day 18, the intra-group difference in tumor volume was just marginally insignificant (*p* = 0.089 and 0.053, respectively; Figure 1d). These results demonstrated that Probio-M9 supplementation slowed down the tumor growth.

### 3.2. Probio-M9 Administration Modulated Mouse Gut Microbiota

Intra-group differences in the gut microbiota were analyzed. The alpha diversity of the mouse fecal microbiota was assessed by the Shannon diversity index. The value of Shannon diversity index of mice in the model group was non-significantly lower than that of the control group but significantly lower than that of the probiotic group (*p* = 0.05; Figure 2a). Then, principal coordinate analysis (Bray–Curtis distance) was performed to visualize differences in the gut microbiota structure between groups (Figure 2b), revealing significant differences were observed in the gut microbiota structure between control and probiotic groups (R^2^ = 0.096, *p* = 0.043; Figure 2b), model and probiotic groups (R^2^ = 0.094, *p* = 0.037; Figure 2b), but not between control and model groups (R^2^ = 0.045, *p* = 0.6; Figure 2b). Although some significant intergroup differences were observed in the gut microbiota, they were not drastic, which is consistent with many previously published probiotic intervention trials in murine models.

A total of 119 SGBs were annotated (Appendix A). The three most abundant SGBs were *Alistipes* sp._1 (4.11%), *Prevotella* sp. MGM2 (3.99%), and *Bacteroides acidifaciens* (3.74%) across all groups. To further analyze the effect of probiotic administration on the gut microbiota composition in mice, significant differentially abundant species that were responsive to probiotic treatment were identified. Overall, there were 14 significant differentially abundant species identified between model and probiotic groups (*p* < 0.05 in all cases; Figure 2c and Appendix A). The fecal microbiota of the probiotic group was significantly enriched in seven responsive SGBs (e.g., *Bacteroidales* bacterium 55_9, *Porphyromonadaceae* bacterium_7, *Rikenellaceae* bacterium_2, and members of the *Alistipes* genus) compared with the model group, while an opposite trend was observed in other SGBs, including *Oscillibacter* sp., *Eubacterium* sp. 14-2_2, *Hungatella hathewayi*_1 and so on. In addition, significant differential species were identified between: the control and model groups (e.g., significantly fewer *Muribaculaceae* bacterium Isolate-013 (NCI)_3 and *Helicobacter japonicus* but more *Duncaniella freteri* and *Lactobacillus taiwanensis* in the model group compared with the control group; *p* < 0.05; Appendix A); control and probiotic group (e.g., significantly fewer *Clostridiales* order and *Desulfovibrio* sp. but more *Rikenellaceae* family in the probiotic group compared with the control group; *p* < 0.05; Appendix A). These results showed that some taxa in the mouse gut microbiota could be responsive to Probio-M9intervention.

### 3.3. Probio-M9 Administration Modulated Mouse Serum and Tumor Mass Cytokine Levels

A ProcartaPlex multiplex immunoassay was performed to assess differences in the serum cytokine profiles between groups. Most of the assayed cytokines did not show significant differences between groups (data not shown). However, the serum levels of some of the monitored cytokines were significantly higher in the probiotic group than the model group, including IFN-γ, IL-9, IL-13, and IL-27, while the level of IL-5 was significantly lower (*p* < 0.05 in all cases; Figure 3a). In addition, the serum IL-5 level was significantly higher in the model group compared with the control group, while an opposite trend was observed in IL-27. Probiotic application prevented the fluctuations in serum IL-5 and IL-27 levels (*p* < 0.05 in all cases; Figure 3a). Histochemical detection in the tumor tissues revealed differences in the abundance and expression of IL-9, IL-27, and IL-5RA positive cells between probiotic and model groups, and such differences were consistent with changes in the serum cytokine profile (Figure 3b,c). These results indicated Probio-M9 intake could regulate the host immunity.

### 3.4. Probio-M9 Administration Modulated Mouse Serum Metabolome

To further reveal the physiological responses of mice towards the Probio-M9 intervention, inter-group differences in the serum metabolome were analyzed. Significant differences in the serum metabolome were observed in all three pair-wise comparisons by principal coordinate analysis, illustrated by the distinct group-based clustering patterns on the respective principal coordinate analysis score plot. Symbols representing the serum metabolome of the control and model groups showed obvious group-based clustering trends (Figure 4a; *p* = 0.001), suggesting there was significant difference in the serum metabolome of tumor transplanted mice compared with control mice without tumor transplantation. The serum metabolome of the probiotic group also exhibited significant differences from the control or the model group (*p* = 0.001 in both cases; Figure 4a), suggesting that Probio-M9 administration modulated the serum metabolome of breast cancer mice and control mice.

Partial least squares-discriminant analyses (cut-off level: VIP score > 2, *p* < 0.05) identified a number of differentially abundant serum metabolites, including 27 metabolites between control and model groups, 42 metabolites between control and probiotic groups, and 35 metabolites between probiotic and model groups (Appendix A). Identified features were compared to the blood exposome database, which annotated 21, 27, and 21 metabolite features, respectively (Appendix A). Interestingly, several metabolites (e.g., pyridoxal, nicotinic acid, 3-hydroxybutyric acid, galactonic acid, kynurenine, and glutamine) were enriched in the probiotic group compared with the model group (*p* < 0.05; Figure 4b). On the other hand, some of the compounds known to be associated with murine mammary tumor, e.g., inositol and lactic acid [37], were enriched in the model group compared with the probiotic group (*p* < 0.05; Figure 4b). These results showed that Probio-M9 intervention could be associated with changes in the host serum metabolome structure and composition.

### 3.5. Correlation among Tumor Growth Inhibition, Gut Microbiota, Serum Cytokines and Metabolites

An association analysis was performed to explore the relationship between mouse tumor volume, gut microbiota, serum cytokines and metabolites (Appendix A). Some interesting correlations were observed. For example, the tumor volume correlated negatively with glutamine (*r* = −0.684, *p* = 0.029). There was significant positive correlation between *Dorea* sp. 5-2 and IL-9, IL-13 (*r* > 0.670, *p* < 0.034), while *Alistipes* sp. CHKCI003 correlated negatively with IL-5, nicotinic acid, and galactonic acid (*r* < −0.654, *p* < 0.05 in all cases).

## 4. Discussion

Breast cancer is one of the most common cancers in women. Provided there is a close relationship between the gut microbiota and cancer pathogenesis and that probiotic administration has the ability of restoring a healthy gut microbiota from disease-associated dysbiotic state, this study analyzed the protective effects of administering Probio-M9 against mammary tumor growth in cancer cell-transplanted mice.

Although there is growing evidence supporting that probiotic administration is a promising strategy in cancer treatment, including breast cancer [38,39], inconsistent clinical outcomes have been reported, which could be a result of the probiotic strain-specificity of beneficial functions. This study chose to evaluate the tumor suppressive effect of Probio-M9 because it showed remarkable efficacy in inhibiting colon cancer and synergized therapeutic effects in mammary tumor treatment when applied jointly with immunotherapy in murine models [27,29]. Our study confirmed that Probio-M9 was also effective in inhibiting mammary tumor growth in cancer cell-transplanted mice. Mice in the probiotic group in this study were given intragastric gavage of probiotics seven days prior to the procedure of tumor transplantation (day −7), and a significant difference in tumor volume between the probiotic and model groups was only observed after 16 days of tumor transplantation, though a non-significant but an obvious trend of difference in tumor volume was already observed at day 14 (*p* = 0.089). At day 12 of tumor transplantation, an obvious tumor lump was not detected in a similar number of mice in the model group (two mice) and probiotic group (three mice). These results suggested that the probiotic intervention was effective in suppressing mammary tumor growth but less likely its formation. It was also possible that a 7-day pre-tumor transplantation probiotic administration was not long enough for developing a significant cancer prophylactic effect, as it did require a moderate duration of daily administration of probiotic (over three weeks from the first probiotic application) to see a significant tumor inhibitory effect. To identify potential anti-tumor mechanisms, we then analyzed differences in the gut microbiome, cytokine profile, and serum metabolome between the model and probiotic groups.

Our microbial metagenomic sequencing and in-depth bioinformatics analysis revealed that the Shannon diversity index decreased in the model group, though the drop was insignificant. However, it is generally thought that a higher gut microbial diversity is beneficial to the host. Moreover, Goedert et al. (2015) reported that postmenopausal women with breast cancer have altered composition and a significantly lower alpha diversity in their fecal microbiota compared with control women [13]. Viaud et al. (2013) demonstrated that the reduction in microbiota diversity by antibiotic treatment could render the mice less responsive to chemotherapy, supporting that a healthy and intact gut microbiota was important for eliciting effective anticancer immune responses [40]. Our observation supported that probiotic application could restore the gut microbiota diversity of tumor-transplanted-mice to a level similar to that of the control group, which is desirable.

Our results of principal coordinate analysis showed that the gut microbiota structure of the probiotic group differed from that of the model group, suggesting that probiotic administration was related to changes in the gut microbiota structure of tumor-bearing mice. Ranjbar et al. (2021) described dysregulated changes in the gut microbiota composition and function through immune- and estrogen-mediated pathways in breast cancer subjects, and concluded that the gut microbiota plays a major role in the development of breast cancer [41]. Moreover, the study of Plaza- Díaz et al. (2019) concluded that women with breast cancer had obviously different microbiota pattern compared with healthy individuals, not only in terms of taxonomic diversity and distribution but also their encoded functionality, such as metabolic capacity and DNA repairing capacity [42]. For example, Luu et al. (2017) found that the abundance of *Bacteroidetes*, *Clostridum coccides*, *Faecalibacterium prausnitzii*, and *Blautia* sp. increased significantly in breast cancer patients and that these taxa might be involved in estrogen, phytoestrogen, and/or SCFA metabolism associating with clinical manifestations [43]. Our results showed the fecal microbiota of the probiotic group had significantly lower abundance in some common SCFA-producing genera, including *Dorea* sp. 5-2, *Roseburia* sp. 1XD42-69, *Eubacterium* sp. 14-2, and *Lachnospiraceae* bacterium A4_2 [44,45,46,47]. On the other hand, *Alistipes* sp._2, *Alistipes* sp._3 and *Rikenellaceae* bacterium_2 were enriched in the fecal microbiota of the probiotic group. Indeed, *Alistipes* is a genus of the *Rikenellaceae* family, and it is also a known SCFA-producing genus in the gut [48]. An increased abundance of *Alistipes* was observed in probiotic-treated hepatocellular carcinoma-bearing mice; this genus could exert tumor suppressive effect via producing anti-inflammatory metabolites together with other gut microbes [49], and it protected against conditions such as colitis and liver fibrosis [50]. Thus, probiotics, as gut microbiota modulators, can be used as tools to harness the host gut microbiome towards increased resistance to tumor growth.

The interactions between host and the gut microbiota are bidirectional, and one commonly adopted effective mechanism of the gut microbiota is producing metabolites such as SCFAs, especially propionate and butyrate, which play critical roles in host homeostasis [51]. It has been shown that SCFAs have an impact on the specific population and function of innate immune cells, particularly on monocytes, macrophages, and natural killer cells, subsequently regulating the host immunity, the balance of Th/Treg cells and pro-/anti-inflammatory cytokines, and the immune responses in tumor therapy [52]. Although our study found varying trends of individual SCFA-producing genera subjected to probiotic treatment, the observed significant tumor-suppressive effect suggested that the overall trend of gut microbiota modulation was beneficial, and that the exact function of each microbe and its role as part of the gut microbiota community merit further elucidation.

In addition, significantly fewer *Oscillibacter* sp., *Hungatella hathewayi*_1, and *Eubacterium* sp. 14-2_2 were detected in the fecal microbiota of the probiotic group compared with the model group. Although these microbes are part of the normal gut microbiota, they might act as opportunistic pathogens under suboptimal conditions. *Oscillibacter*-like organisms are thought to be associated with high-fat-diet-induced gut dysfunction, possibly via disrupting the gut barrier integrity in the proximal colon [53]; *Hungatella hathewayi* has been found to advance intestinal tumorigenesis via regulating multiple tumor suppressor gene promoters on the epigenetic level [54]; *Eubacterium rectale* could initiate colorectal cancer through promoting colon inflammation [55]. Therefore, one possible mechanism of the tumor suppressor effect of Probio-M9 is inhibiting certain opportunistic microbes in the gut environment in mammary tumor-bearing mice.

A rich and diverse gut microbiota is a prerequisite for healthy development and maturation of the host immune system [56], and cytokines have been shown to regulate estrogen synthesis in breast tumors [57]. In fact, there is evidence supporting that the protective effect of probiotics (including *Lactobacillus helveticus*, *Limosilactobacillus reuteri*, *Lactiplantibacillus plantarum*, and *Lacticaseibacillus casei* Shirota) against breast cancer progression is via modulating the host immune system [24,25,58,59]. We thus assessed the effect of Probio-M9 on the immunity in our tumor-transplanted mice. It was observed that the serum IFN-γ concentration in the probiotic group was significantly higher than that of the model group. Interferon-γ plays a direct role in inhibiting proliferation and promoting apoptosis of tumor cells. A previous study reported that administering *Lactobacillus acidophilus* increased the IFN-γ level in an induced murine breast cancer model [60]. Moreover, a Th1/Th2 imbalance biased towards Th2 response was observed in breast cancer patients, and Th1 dominant response increased patients’ survival rate while decreasing the chance of cancer reoccurrence [61]. Our study found that probiotic administration shifted the Th1/Th2 balance more towards a Th1 response, characterized by a decreased serum IL-5 level but an increase in IFN-γ, though increases in some pleiotropic cytokines (IL-9, IL13, and IL-27) were observed simultaneously. Both IL-9 and IL-27 are pleiotropic cytokines that could induce innate/adaptive immune responses and promote tumor cell apoptosis by enhancing the action of cytotoxic T lymphocytes [62,63]. The serum levels of both IL-9 and IL-27 were significantly higher in the probiotic group compared with the model group, which potentially played a role in inhibiting tumor growth. Fang et al. (2015) found that IL-9 strongly inhibited the growth of two melanoma cell lines, namely HTB-72 and SK-Mel-5, and induced apoptosis of HTB-72 cells [64]. However, our results notably showed that the probiotic group had also a higher level of serum IL-13 compared with the model group; IL-13 is another pleiotropic cytokine involved in negative regulation of anti-tumor immunity [65]. The reason for the higher level of IL-13 after probiotic administration in the current model merits further investigation. These results together suggested that Probio-M9 could enhance the immune function in mice-bearing mammary tumor and shift more towards a Th1 response.

The alteration of the serum metabolome in cancer patients has been a focus in many studies [66]. Probiotic-driven changes in the gut microbiota composition would expectedly result in alterations in the serum metabolome. The serum metabolome of Probio-M9-administered mice had significantly higher levels of nicotinic acid (a common form of vitamin B3), pyridoxamine (vitamin B6), and amino acid derivatives (including glutamine, kynurenine and 3-hydroxybutyric acid) compared with the model group. Both vitamin B3 and B6 are beneficial for gut health. Vitamin B3 could lessen inflammation, participate in maintaining genomic stability, counteract tumorigenesis, and, possibly, reduce cancer risk [67,68]. Vitamin B6 was found to affect the cell cycle, inflammation, angiogenesis, oxidative stress, and chromosomal stability; however, inconsistent data exist regarding the role of vitamin B6 in cancer protection. Epidemiological evidence suggests that vitamin B could be a potential risk reduction agent, while data from randomized clinical trials did not show convincing cancer protective effects [69].

Amino acids are responsible for a multitude of essential functions within the human body, including homeostasis, biosynthesis, energetic regulation, redox balance, and cancer metabolism [70]. The gut microbiota plays a key role in intestinal protein/amino acid metabolism. The probiotic group had significantly higher serum levels of glutamine and β-hydroxybutyric acid compared with the model group. β-Hydroxybutyric is the most common ketone in the human body that could inhibit inflammation, lipid metabolism, and regulate the gut microbiota through various mechanisms and signaling molecules [71]. Mitigation of metabolic dysregulation by depriving the glucose availability to tumor cells is a novel cancer therapeutic strategy; thus, dietary supplementation of a high-fat, low-carbohydrate ketogenic diet, e.g., β-hydroxybutyric, might serve such purpose by elevating blood ketones in place of sugars to fulfill the energy demand of normal tissues. Preclinical studies have demonstrated promising anti-tumor effects of a ketogenic diet [72]. For example, Zou et al. (2020) demonstrated remarkable inhibitory effects of a ketogenic diet with/without co-administering rapamycin against tumor growth and lung metastasis in a murine breast cancer model [73].

Significant negative correlation was found between the serum glutamine level and mouse, tumor volume; glutamine has been shown involvement in a variety of non-anabolic cellular functions, e.g., regulating cell survival, promoting enterocyte proliferation, conferring oxidative stress resistance, and suppressing proinflammatory signaling pathways [74,75]. Glutamine could inhibit tumor growth through enhancing the immune function (e.g., increased natural killer cell activity after oral intake to suppress breast tumor growth in rats) and protect the host by raising the tumor selectivity in radiation or chemotherapy [76]. We speculate that the observed changes in specific serum metabolites represent probiotic-driven host physiological responses against tumor growth.

This work has some limitations. Firstly, the inclusion of a probiotic control without tumor transplantation could improve the trial design, as it would provide information of adverse effect of the use of Probio-M9, a human-originated probiotic. However, based on our previous and unpublished works [29], we already knew that Probio-M9 would not cause undesirable effect to the mice, so such control group was not included here. Secondly, we did not observe Probio-M9 colonization in the probiotic group. Tracking the changes in the ingested strain in the host gut and identifying the association between the bacterial dynamics and the beneficial effect would provide insights into the probiotic function. However, this is not always possible, as the quantity of the ingested probiotics is only a small proportion relative to the complete host gut microbiota, especially with a sequencing depth of routine metagenomic sequencing studies such as the current one (i.e., ~5.5 ± 1.49 Gbp raw data per sample). Indeed, we did detect some lactobacilli in the mouse gut microbiota (e.g., *Lactobacillus murinus* and *Lactobacillus taiwanensis*, 0.97% and 0.94% on the SGB level, respectively, but not *Lacticaseibacillus rhamnosis*), and the detected lactobacilli are likely part of their endogenous gut microbiota. The failure in detecting *Lacticaseibacillus rhamnosis*/Probio-M8 could reflect that this species/strain was truly absent in the samples or that the concentration of probiotics was too low to be detected after being ingested and through the gastrointestinal transit. In our case, we tend to believe that it is the latter reason, as, generally, at least fivefold sequencing coverage is required for tracking a bacterial strain. So, it is not surprising that the current sequencing depth is inadequate for detecting the ingested probiotics. Moreover, the fact that we failed to detect *Lacticaseibacillus rhamnosis* (of which Probio-M9 belongs to) in our samples reflects that this species is probably not a major naturally existing microbe in the mouse gut, so it is not logical to anticipate that the Probio-M9 could colonize and expand in the mouse gut by a large magnitude. Although they are not detectable under the current condition and by typical metagenomic sequencing, it is known that probiotics are still able to confer beneficial effects to the host as allochthonous gut microbiota. Thirdly, although the model and probiotic groups showed significant differences in tumor volume, a high intragroup variability was observed. Moreover, the intergroup differences in the gut microbiome and metabolome were modest, though significant differences were detected by multivariate analysis such as the adonis test. These results support that Probio-M9 exerted some degree of tumor growth suppression. It would, however, be necessary to confirm the current findings and elucidate the probiotic mechanism in future animal trials with a larger sample size.

In conclusion, the current mouse model serves as a pilot study showing the beneficial effect of administering a probiotic, *Lacticaseibacillus rhamnosus* Probio-M9, in slowing the growth of transplanted mammary tumor, and such effect was accompanied by a multitude of host gut microbiota, immune, and metabolic responses (Figure 5). Larger scale researches and human clinical studies are still required to further validate the current findings.

## Figures and Tables

**Figure 1 nutrients-15-00005-f001:**
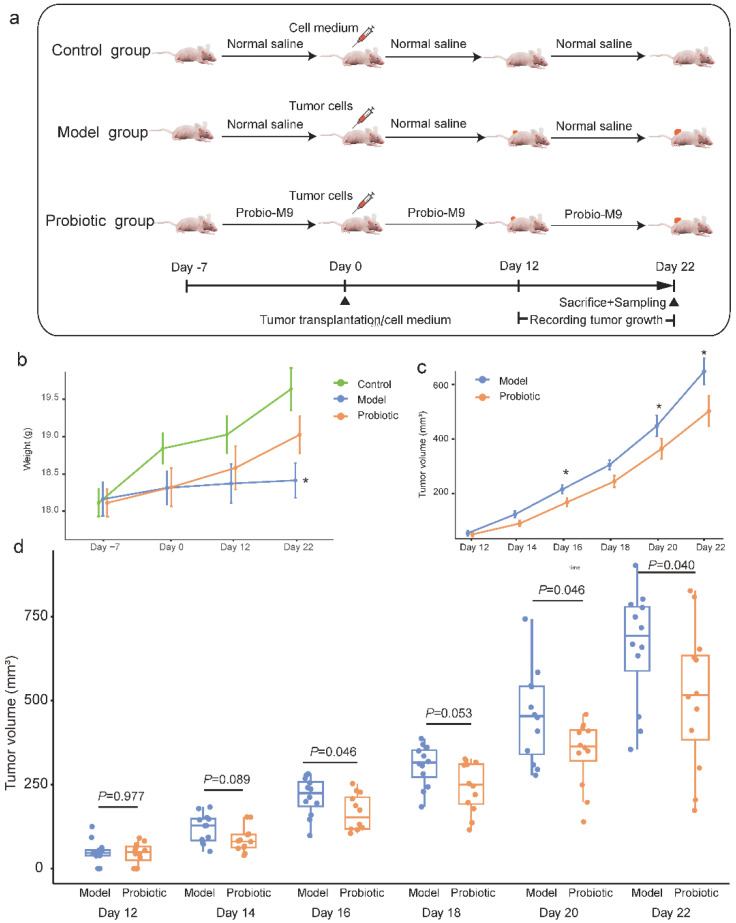
Experimental design and changes in tumor volume in mice. (**a**) A total of 36 mice were randomized into the control, model, and probiotic groups (*n* = 12 per group), respectively, for different interventions. Tumor cells were transplanted to the mice in the model and probiotic groups at day 0. Tumor growth was monitored every other day between day 12 and day 22. At day 22, mice were sacrificed, and tumor tissue, fecal and blood samples were collected. (**b**) The line chart shows the changes in the body weight of mice over time. Error bars represent the standard error of the mean. * *p* < 0.05 (model group versus probiotic group), Wilcoxon test. (**c**) The line chart shows average changes in tumor volume of mice over time. Error bars represent the standard error of the mean. The asterisk represents significant intra-group difference, * *p* < 0.05, Wilcoxon test. (**d**) The box plot shows changes in the tumor volume during the course of intervention. The *p* value of each pairwise intra-group comparison was calculated by Wilcoxon test.

**Figure 2 nutrients-15-00005-f002:**
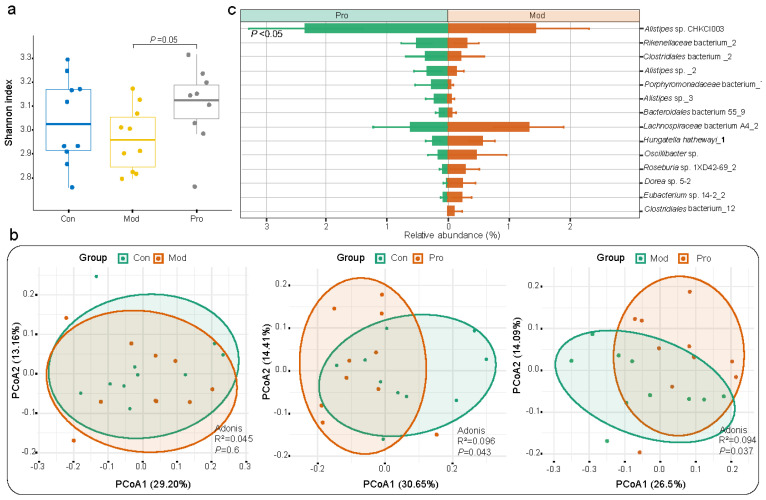
Microbial diversity and species−level genome bin (SGB) features of fecal metagenome dataset of mice. (**a**) Shannon diversity index of control (Con), model (Mod), and probiotic (Pro) groups at day 22. (**b**) Principal coordinates analysis (PCoA; Bray−Curtis distance) score plots of fecal microbiota. Symbols representing samples of the two groups are shown in different colors. (**c**) Responsive SGBs showing significant differential abundance between probiotic and model groups at day 22 (*p* < 0.05; Wilcoxon test).

**Figure 3 nutrients-15-00005-f003:**
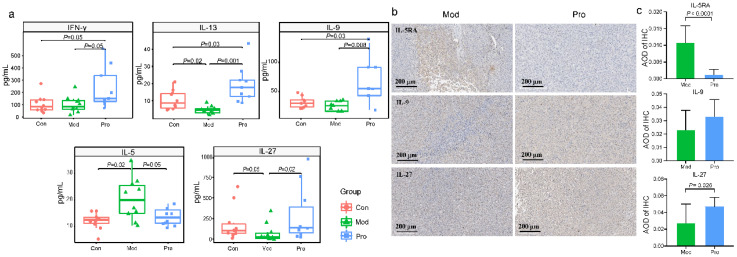
Serum cytokine levels and histochemical staining of cytokines in tumor tissues of the control (Con), model (Mod), and probiotic (Pro) groups at day 22. (**a**) Differences in serum concentrations of interferon-γ (IFN-γ), interleukin (IL)-13, IL-9, IL-5, and IL-27 between groups were evaluated using Wilcoxon tests. (**b**) Immunohistochemistry (IHC) of IL-5RA, IL-9, and IL-27 of representative sections of tumor tissues of Mod and Pro groups, 25× magnification (scale bar represents 200 μm). (**c**) the average optical density (AOD) of IHC between Mod and Pro groups were evaluated using Wilcoxon tests.

**Figure 4 nutrients-15-00005-f004:**
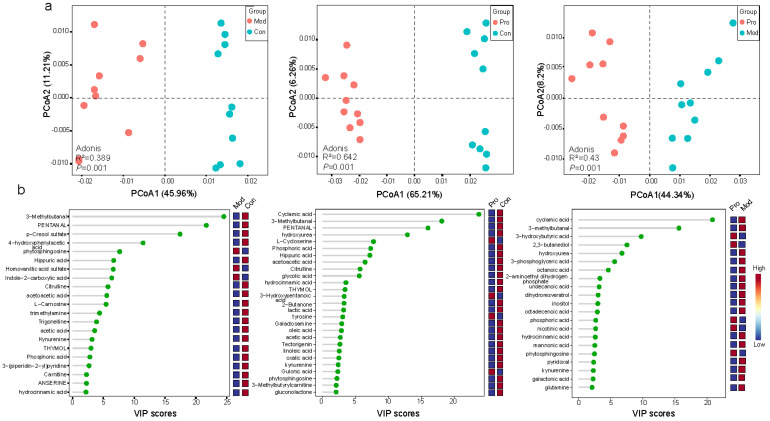
Comparison between the serum metabolome of the control (Con), model (Mod), and probiotic (Pro) groups at day 22. (**a**) Principal coordinates analysis (PCoA; Bray-Curtis distance) score plots of the serum metabolome between groups. (**b**) Annotated significant differential metabolites: Con versus Mod groups; Con versus Pro groups; Mod versus Pro groups, detected by liquid chromatography-mass spectrophotometry. Cut-off threshold: variable importance in projection (VIP) score > 2 and *p* < 0.05.

**Figure 5 nutrients-15-00005-f005:**
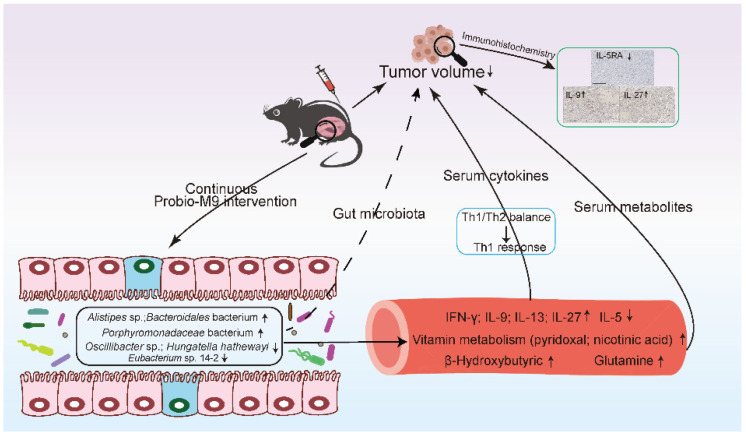
Proposed mechanism of *Lacticaseibacillus rhamnosus* Probio-M9 in inhibiting mammary tumor growth.

## Data Availability

Not applicable.

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
