# Peer review of "Lacticaseibacillus rhamnosus Probio-M9-Driven Mouse Mammary Tumor-Inhibitory Effect Is Accompanied by Modulation of Host Gut Microbiota, Immunity, and Serum Metabolome"

_nutrients, 2022, doi:10.3390/nu15010005_

Round 1
Reviewer 1 Report
Zhang et al. investigated the anti-tumorigenesis potential of the Probio-M9, a novel probiotic strain isolated from healthy human breast milk and has been shown to suppress colitis-associated tumorigenesis, on mammary cancer in a tumor transplantation murine model. This well-written manuscript provided innovative proof that oral administration of Probio-9 could modulate host immunity offering a promising tumor-inhibitory effect.
Comments:
1. A major concern is that the confirmation of Probio-M9 colonization in the Probiotic group is missing. In general, oral administration of human gut microbiota in mice models requires approximately 4-6 weeks to reach stable colonization. The procedure of 30-day oral gavage of Probio-M9 and their colonization needs to be assessed.
2. A critical control group with the same procedure of 30-day oral gavage of Probio-M9 is missing. Longitudinal vital signs of the mice, for instance, body weight, should be monitored when human microbiota has been introduced to the BALB/c nude mice.
3. The authors stated that breast cancer, as estrogen promotes the proliferation of normal breast epithelium and cancer cells. Whether the estrogen level in the BALB/c nude had been altered due to the tumor transplantation in the Model group? What is the effect of Probio-M9 on estrogen levels in the Probiotic group mice?
4. Introduction section, regarding the hypothesis that “Probio-M9 could also suppress mammary tumor growth….” Suggest building exclusive associations between breast cancer and Proioa-M9. Why specifically focus on breast cancer needs to be clarified.
5. Fig 3b is unclear, and it is hard to appreciate cellular morphology. Quantification of the signals?
Author Response
Response to Reviewer 1 Comments
Point 1: A major concern is that the confirmation of Probio-M9 colonization in the Probiotic group is missing. In general, oral administration of human gut microbiota in mice models requires approximately 4-6 weeks to reach stable colonization. The procedure of 30-day oral gavage of Probio-M9 and their colonization needs to be assessed.
Response 1: Thank you for your comment. We agree with this Reviewer that it would be interesting to track the changes in the ingested probiotic strain in the gut of the mice. Particularly, tracking the changes in the ingested strain in the host gut would provide insights into the aspect of gut colonization.
However, this is not always possible as the quantity of ingested probiotics is only a small proportion relative to the complete gut microbiota in the host, especially with a sequencing depth of routine metagenomic sequencing studies like the current one (i.e., ~ 5.5 ± 1.49 Gbp raw data per sample). Our taxonomic analysis did detect some lactobacilli (e.g., Lactobacillus murinus, Lactobacillus taiwanensis; 0.97% and 0.94% on the SGB level, respectively; Table S1), which we believe are the endogenous and major intestinal lactobacilli in the mouse gut. We did not detect any Lacticaseibacillus rhamnosis (of which the ingested probiotic, Probio-M9, belongs to). The negative result could either reflect that the ingested probiotics were absent in the samples or that the cell density of the ingested probiotics was too low to be detected after the gastrointestinal transit. In our case, we tend to believe that it is the latter reason, as, generally, at least 5-fold sequencing coverage is required for tracing a bacterial strain according to our experience in metagenomic analysis and strain-level bacterial tracking. So, we have to admit that the sequencing coverage was likely inadequate for tracking the ingested strain. Considering the cost, it is not always possible to apply ultra-deep sequencing. Moreover, as the species Lacticaseibacillus rhamnosis was not detected in our samples, it reflects that this species is probably not a major naturally existing endogenous microbe in mouse gut, so it is not logical to anticipate that the ingested strain could colonize and expand in mouse gut in a large magnitude. Although they are not detectable under the current condition and by typical metagenomic sequencing, probiotics are still able to confer beneficial effects to the host even as allochthonous gut microbiota.
In regard to the length of the trial, as we used nude mice in this work, the implanted tumor grew rather quickly, impacting the daily activities and health/physiological state of the tumor-bearing mice as the trial continued. If the time of trial was further extended, there would be concerns about animal welfare and ethics, so this trial was designed as it is.
After all, the reviewer raised an interesting aspect of the study, which should be specifically addressed when planning future studies (probably by using an ultradeep metagenomic sequencing approach and with an exception high level of probiotic administration). We are very grateful for this insightful suggestion, and we have added some of the above content in the Discussion section.
Point 2: A critical control group with the same procedure of 30-day oral gavage of Probio-M9 is missing. Longitudinal vital signs of the mice, for instance, body weight, should be monitored when human microbiota has been introduced to the BALB/c nude mice
Response 2:Thank you for your comment. We agree with this Reviewer that the trial design would have been more complete if a control group of 30-day oral gavage of Probio-M9 could have been included, and this is especially important to show that Probio-M9 administration did not exert any adverse effect on the mice. We acknowledge this limitation in the Discussion.
Indeed, we did monitor the changes in body weight of the three groups of mice, and the data are now described (line 243-249 in the revised manuscript; Figure 1b). Based on our preliminary experiments and previous works, we already knew that this bacterium did not cause any adverse effect in nude mice and tumor-induced mice (Gao et al., 2021, Adjunctive Probiotic Lactobacillus rhamnosus Probio-M9 Administration Enhances the Effect of Anti-PD-1 Antitumor Therapy via Restoring Antibiotic-Disrupted Gut Microbiota. Front. Immunol. 12:772532; Xu et al., 2021, Inhibitory Effects of Breast Milk-Derived Lactobacillus rhamnosus Probio-M9 on Colitis-Associated Carcinogenesis by Restoration of the Gut Microbiota in a Mouse Model. Nutrients 13:1143).
Point 3: The authors stated that breast cancer, as estrogen promotes the proliferation of normal breast epithelium and cancer cells. Whether the estrogen level in the BALB/c nude had been altered due to the tumor transplantation in the Model group? What is the effect of Probio-M9 on estrogen levels in the Probiotic group mice?
Response 3: Thank you for the interesting question. It is a fact that estrogen is related to cancer development and growth. However, in this work, we did not focus on the endocrine system and did not monitor changes in the estrogen level in the experimental mice. So, to be very honest, we do not know the answer to your question.
We think that this is a very interesting question, so we searched in the Pubmed database with the key words “Lactobacillus rhamnosis estrogen”, returning only 24 results. None of the returned reports investigated the effect of L. rhamnosis application on estrogen in a cancer model. So, this is indeed an area that needs and would be of interest to explore in future. Particularly, provided the strain specificity of probiotic effects and their individualized host health impact, your intriguing question certainly merits future investigation.
Point 4: Introduction section, regarding the hypothesis that “Probio-M9 could also suppress mammary tumor growth….” Suggest building exclusive associations between breast cancer and Proioa-M9. Why specifically focus on breast cancer needs to be clarified.
Response 4: In our unpublished study (manuscript under review), we demonstrated that the probiotic Probio-M9 could be transferred from the intestine to the mammary gland, suggesting the existence of an entero-mammary pathway of probiotic translocation. In addition, Probio-M9 could prevent, alleviate and adjuvant antibiotic treatment of bacterial mastitis in lactating rats after intestinal-mammary migration (manuscript under review). In another study, we found that Probio-M9 administration could enhance the efficacy and responsiveness of anti-PD-1-based immunotherapy, and Probio-M9 could be a potential candidate of microbe-based synergistic tumor therapeutics (Gao et al., 2021, Adjunctive Probiotic Lactobacillus rhamnosus Probio-M9 Administration Enhances the Effect of Anti-PD-1 Antitumor Therapy via Restoring Antibiotic-Disrupted Gut Microbiota. Front. Immunol. 12:772532).
The results of our previous works together suggested that this strain could translocate to the mammary gland and exert localized beneficial effect. Thus, we chose to focus on the beneficial effect of Probio-M9 in breast cancer in this study. We have added more information to link “breast cancer” and Probio-M9 in the last paragraphs in the Introduction section.
Point 5: Fig 3b is unclear, and it is hard to appreciate cellular morphology. Quantification of the signals?
Response 5:Thank you for your comment. We have zoomed in on the histological sections (From 10x to 25x) and quantified the signals. Please see Figure 3b, 3c in the updated manuscript.

Reviewer 2 Report
In this manuscript, the authors analyze the effect of Pro-bio M9 administration in a murine model of breast cancer. The authors evaluate and compare four parameters: 1-tumor size, 2-fecal bacterial diversity, 3-cytokine profile, 4-metabolomics profile in three groups: a control group (no cancer cells and no ProbioM9), a model group (cancer cells but no ProbioM9) and a treatment group (cancer cells + ProbioM9). The paper is well written and correctly detailed; however, some points need to be clarified.
Major comments: Although one can understand the authors' enthusiasm for the work, their conclusions should objectively reflect the observation and in some cases, they go beyond it.
Line 253-255, the authors claim: “These results demonstrated that Probio-M9 supplementation significantly decreased the tumor growth rate”. However, Figure 1 shows that tumor growth appears to be slightly slower in the treated group but the growth rate appears to be the same, so analysis of tumor size at day 24 would be more appropriate.
Line 274, the authors claim: “These results suggested that tumor transplantation mildly reduced the alpha diversity of gut fecal microbiota, but continuous probiotic intake reversed such effect.” “Probiotic intervention also significantly modulated the fecal microbiota structure of the mice” However this cannot be concluded by the results shown in figure 2b, for two main reasons: 1- there is no control group with only the probiotic and 2-the model and control appear very similar
Line 295 and fig 2c. , the authors claim: “These results showed that Probio-M9 regulated the mouse gut microbiota composition”. However, Figure 2c shows the abundance of 14 species, 7 are more abundant in the Pro group and the other 7 in the Mod group (please see above); In addition, I think that figure 2c is confusing, maybe with the bars on the same side would be clearer.
Line 335, the authors claim: “suggesting that Probio-M9 administration modulated the overall serum metabolome of breast cancer mice and control mice” However the overall metabolome was not analyzed and there is no control group with only the probiotic
Line 532 the authors claim: “In conclusion, this study has provided proof of principal that oral supplementation of probiotics like Lacticaseibacillus rhamnosus Probio-M9 could inhibit mammary tumor growth in mice” this conclusion is far from the results shown, please see previous comments.
Author Response
Response to Reviewer 2 Comments
Point 1: Line 253-255, the authors claim: “These results demonstrated that Probio-M9 supplementation significantly decreased the tumor growth rate”. However, Figure 1 shows that tumor growth appears to be slightly slower in the treated group but the growth rate appears to be the same, so analysis of tumor size at day 24 would be more appropriate.
Response 1: Thank you for your comment. Since the trial ended at day 22, it was not possible to measure the size of the tumor at day 24.
In Figure 1b, we can see that the slope of the tumor size change between days 20 to 22 was deeper in the model group than the probiotic group. Moreover, the intergroup differences in tumor volume at both days 20 and 22 showed significant differences (probiotic group < model group, P < 0.05 at both time points). These two observations supported that Probio-M9 administration decreased the tumor growth. One potential and related concern mentioned by Reviewer 3 is the high intragroup variability though the statistical analysis returned a significant difference in the tumor size. We acknowledge that these findings should be validated in future works with a larger sample size (the current sample size is n = 12 per group, which is comparable to works of similar nature).
We have added Discussion in this regard, and we clarified the description by not emphasizing the differences in “tumor growth rate” modified the statement to: “These results demonstrated that Probio-M9 supplementation slowed down the tumor growth.” (Line 265-266).
Point 2:Line 274, the authors claim: “These results suggested that tumor transplantation mildly reduced the alpha diversity of gut fecal microbiota, but continuous probiotic intake reversed such effect.” “Probiotic intervention also significantly modulated the fecal microbiota structure of the mice” However this cannot be concluded by the results shown in figure 2b, for two main reasons: 1- there is no control group with only the probiotic and 2-the model and control appear very similar.
Response 2: Thank you for your comment. We agree with this Reviewer that it would be more complete if a probiotic-no tumor control group could have been included, especially to find out whether the probiotics caused adverse effect to mice. Indeed, based on our preliminary experiments and previous works, we already knew that this bacterium did not cause any adverse effect in nude mice and tumor-induced mice (Gao et al., 2021, Adjunctive Probiotic Lactobacillus rhamnosus Probio-M9 Administration Enhances the Effect of Anti-PD-1 Antitumor Therapy via Restoring Antibiotic-Disrupted Gut Microbiota. Front. Immunol. 12:772532; Xu et al., 2021, Inhibitory Effects of Breast Milk-Derived Lactobacillus rhamnosus Probio-M9 on Colitis-Associated Carcinogenesis by Restoration of the Gut Microbiota in a Mouse Model. Nutrients 13:1143).
As the aim of this work was to investigate the effect of probiotic administration on tumor-bearing mice (tumor growth, gut microbiota etc.) For this purpose, the probiotic only group becomes less important. We do thank this Reviewer for the suggestion and will consider about this in future trial design. We also acknowledged this limitation in the Discussion section.
We do agree with the Reviewer that the differences are not drastic. Indeed, according to currently available publications, probiotic-mediated gut microbiota changes are usually mild to modest, similar to the results seen in this study. In our study, the results in Figure 2b are supported by statistical analysis. The difference in Shannon diversity between the model and probiotic group was supported by Wilcoxon test (P = 0.05). Then, principal coordinate analysis of the gut microbiota structure was performed by pairwise comparison. Although some overlapping was observed on the PCoA score plots, the results of Adonis test did suggest some differences existed between groups (control versus probiotic group, P = 0.043; model versus probiotic group, P = 0.037). For example, for model versus probiotic group (right panel of Figure 2b) symbols representing the model group are mainly located at the lower part and upper left of the score plot, while those representing the probiotic group are mainly located at the upper right side of the score plot. We have now included the results of the adonis tests in the PCoA score plots.
We do agree with this Reviewer’s comments, so we toned down the inferences drawn at the end of this subsection by replacing the two mentioned statements with: “Although some significant intergroup differences were observed in the gut microbiota, they were not drastic, which is consistent with many previously published probiotic intervention trials in murine models.” (line 277-279).
Point 3: Line 295 and fig 2c., the authors claim: “These results showed that Probio-M9 regulated the mouse gut microbiota composition”. However, Figure 2c shows the abundance of 14 species, 7 are more abundant in the Pro group and the other 7 in the Mod group (please see above); In addition, I think that figure 2c is confusing, maybe with the bars on the same side would be clearer.
Response 3: Thank you for your comment. We clarified the inference of this subsection: “These results showed that some taxa in the mouse gut microbiota seemed to be responsive to Probio-M9 intervention”. We checked Figure 2c and agreed with the Reviewer that the original figure was confusing. So, we modified if by rearranging the sequence of taxa based on their relative abundance. It looks neat and clearer now. Thank you again for your suggestion.
Point 4:Line 335, the authors claim: “suggesting that Probio-M9 administration modulated the overall serum metabolome of breast cancer mice and control mice” However the overall metabolome was not analyzed and there is no control group with only the probiotic
Response 4: Thank you for your comment. The overall serum metabolome was compared in pair by principal coordinate analysis, which is a dimension reduction method to explore and to visualize similarities or dissimilarities of datasets. We did observe significant intergroup differences in the serum metabolomes on the respective PCoA score plots (Figures 4a, 4b, 4c; adonis test, P = 0.01 in all three cases). We do agree with this Reviewer that the trial design would be more complete if a probiotic control group was included. With the current data, we could only compare between the metabolome of the model group and control group, showing that there was difference in the serum metabolome between tumor-bearing mice and those without; between the metabolome of the probiotic group and control group to show that Probio-M9 administration could modulate the serum metabolome; between the metabolome of the probiotic group and model group to show that Probio-M9 could partly reverse metabolite changes due to tumor transplantation.
We have acknowledged the limitation of not including a probiotic only group in the Discussion section, which should be included in future experimental design. Also, to tone down our inference, we removed the word “overall” in the statement. Thank you again for your comment.
Point 5:Line 532 the authors claim: “In conclusion, this study has provided proof of principal that oral supplementation of probiotics like Lacticaseibacillus rhamnosus Probio-M9 could inhibit mammary tumor growth in mice” this conclusion is far from the results shown, please see previous comments.
Response 5: Thank you for your comment. We agree with this Reviewer that this work in itself is not enough, particularly because of the high intragroup variability. We have toned down this conclusion, and we specifically stated that it was a pilot study and larger scale studies should be conducted in future to further validate the current observations. Moreover, in the Discussion section, we have listed the limitations of this study, so that readers are informed.

Reviewer 3 Report
Review comments:
Short summary: Zhang et al., investigated the suppressive effect of Lacticaseibacillus rhamnosus Probio-M9 on tumor growth in mice transplanted with mammary cancer cells. They found a modulative effect of Probio-M9 in host gut microbiome, immunity, and metabolism. Finally, the authors provide an interesting list of correlations between microbiome, immunity, and metabolism. Their work shows a proof of principal indicating probiotic administration as both preventive and therapeutic effect in slowing the growth of transplanted mammary tumor in mice.
Strength of the study:
Although a similar study has been published 12 years ago (Yazdi et al., Br J Nutr 2010), this study has several novel aspects. The presentation and the scientific soundness of the paper is of good quality and merits publication. The proposed mechanism shown in Figure 5 is important and gives a pictorial view of the research outcome.
While this research is relevant, there are few concerns, comments and questions that needs clarification:
· In the trial design of the methodology, a very high dose of Probio-M9 is stated in Line 143. What is the rational for using such a high dose/day, which is similar to the typical dose in human supplements? In any case it would be relevant to report or indicate any toxic side effect of the Probio-M9 administered or observed during the study.
· In Figure 1, from day 20 to day 22, there is a high variability in the graph though significant difference is seen between the model and the probiotic groups. Was there any physical changes/signs observed with the mice at this stage?
· Was there any signs of weight loss or gain after injecting mice with the cancer cells from start to end of experiment? Was there any death of mice? It would be of scientific relevance to include these, if any?
· Line 314, please clarify whether IL-5R or IL-5 expression was analyzed, since the descriptions in the text and the legend for Figure 3 all indicate a measurement for IL-5.
· Kindly include any limitation of the study in the discussion.
· Please provide human readable names for the metabolites in Figure 4d. Please also provide the full list of significantly different metabolites between all groups as supplementary data.
· Please increase the zoom for the histology sections (Figure 3b).
· Figure 5: The authors show good correlations between microbiome and cytokines and metabolites. Accordingly, there should be an arrow from microbiome to vessel. On the other hand, the correlation from microbiome to tumor volume is probably not direct but an indirect effect transmitted by cytokines and metabolites. Accordingly, the direct arrow from gut microbiota to tumor volume might be removed.
· Please provide original histology images and all measures of tumor length/width/volume, IFN/IL13/IL9/IL5/IL27 in a spread sheet for review purposes only.
· Supp table S1, why do some species show up more than once, e.g., Lachnospiraceae bacterium A4 in lines 18, 20, 21.
Would it make sense to include more species?
Please at least include all species that are included in table S2.
Why do some species from table S2 with abundances > 1% not show up in table S1? E.g., Lactobacillus taiwanensis has abundance 1.17 in model, but is not included in Table S1.
· Supp table S2, why do some species show up more than once in each comparison? E.g., “Clostridiales bacterium” in Control group vs Probiotic group, lines 14 and 21, and “Rikenellaceae bacterium” in lines 26 and 28.
What do the mean values indicate, is it relative abundance?
Minor comments:
Font size in the abstract is inconsistent (check from Line 27 to 30)
Line 108, should read “a“ not “an“
Line 523, delete “in“
Final decision:
Accept after addressing comments
Author Response
Response to Reviewer 3 Comments
Point 1: In Figure 1, from day 20 to day 22, there is a high variability in the graph though significant difference is seen between the model and the probiotic groups. Was there any physical changes/signs observed with the mice at this stage?
Response 1: Thank you for your comment. Yes, we agree with this Reviewer that there was a rather high variability. At the last days of the trial, the behavioral activity and mental status of the mice in the probiotic group did seem to be better than those in the model group based on our daily observation. However, it was difficult to quantify these aspects in this work. A larger scale work (of an increased sample size) with more resources and supports should be planned in the future to address differences in other health aspects of the animals.
Point 2: Was there any signs of weight loss or gain after injecting mice with the cancer cells from start to end of experiment? Was there any death of mice? It would be of scientific relevance to include these, if any?
Response 2: Thank you for your comment. None of the mice died during the trial. Indeed, we monitored the body weight change in the mice. The data are now included in Figure 1b. Tumor transplantation has obvious but mostly non-significant suppressive effect on the mouse body weight gain, and the only significant difference noted between the model and probiotic group was at the last time point (day 22) after tumor transplant. The results are now described in the manuscript (line 243-249).
Point 3: Line 314, please clarify whether IL-5R or IL-5 expression was analyzed, since the descriptions in the text and the legend for Figure 3 all indicate a measurement for IL-5
Response 3: Thank you for pointing this out. We checked that the multiple cytokine array kit tested IL-5 expression, while IL-5R was analyzed histologically. This is now clarified in the manuscript.
Point 4: Kindly include any limitation of the study in the discussion
Response 4: Thank you for this suggestion. We have to admit that there are many limitations in this study. We have included them in the Discussion section.
Point 5: Please provide human readable names for the metabolites in Figure 4d. Please also provide the full list of significantly different metabolites between all groups as supplementary data.
Response 5: We have provided the full list of significantly different metabolites between groups in Table S6. Figure 4d is modified to provide readable names.
Point 6: Please increase the zoom for the histology sections (Figure 3b).
Response 6: Thank you for your comment. We have zoomed in on the histological sections and quantified the signals (From 10x to 25x; Figure 3b, 3c).
Point 7: Figure 5: The authors show good correlations between microbiome and cytokines and metabolites. Accordingly, there should be an arrow from microbiome to vessel. On the other hand, the correlation from microbiome to tumor volume is probably not direct but an indirect effect transmitted by cytokines and metabolites. Accordingly, the direct arrow from gut microbiota to tumor volume might be removed.
Response 7: We have modified the figure as suggested. Thank you for the suggestions.
Point 8: Please provide original histology images and all measures of tumor length/width/volume, IFN/IL13/IL9/IL5/IL27 in a spread sheet for review purposes only.
Response 8: We have provided the original histology images and the mentioned data in a spread sheet for review.
Point 9: Supp table S1, why do some species show up more than once, e.g., Lachnospiraceae bacterium A4 in lines 18, 20, 21.
Would it make sense to include more species?
Please at least include all species that are included in table S2.
Why do some species from table S2 with abundances > 1% not show up in table S1? E.g., Lactobacillus taiwanensis has abundance 1.17 in model, but is not included in Table S1.
Response 9: We apologize for the confusion. We have rearranged Table S1 and included the complete list of identified SGBs, so that the data are presented more clearly. Also, all differential SGBs listed in Table S2 are now included in Table S1.
The species-level genome bins (SGBs) classify taxa on the species level. Some SGBs appear to have the same name because they were assigned under the same species; however, based on their genome features and the set parameters in the metagenomic analysis pipeline, they were indeed distinctive enough to be separated into different “types” under the same species. In other words, they could be considered as different “strains”. To illustrate their differences, we have numbered them after the species name, accordingly. For example, the different Lachnospiraceae bacterium A4 is now numbered as Lachnospiraceae bacterium A4_1, Lachnospiraceae bacterium A4_2, Lachnospiraceae bacterium A4_3, etc. to indicate that they are indeed different.
Point 10: Supp table S2, why do some species show up more than once in each comparison? E.g., “Clostridiales bacterium” in Control group vs Probiotic group, lines 14 and 21, and “Rikenellaceae bacterium” in lines 26 and 28.
What do the mean values indicate, is it relative abundance?
Response 10: Thank you again for your question. The same reason as above, the repeated names indeed refer to different taxa (strains) under the same sub-species division. Accordingly, we have added a number code after the assigned species name in each case to distinguish between them.
The mean value refers to the relative abundance, which is now specified in the table. Thank you.

Round 2
Reviewer 2 Report
Authors have answered the questions and modified the document accordingly, it is appreciated that they have included the limits of their experiences.